# Black-Box Optimization Revisited: Improving Algorithm Selection Wizards through Massive Benchmarking

## Abstract

Existing studies in black-box optimization for machine learning suffer from low generalizability, caused by a typically selective choice of problem instances used for training and testing different optimization algorithms. Among other issues, this practice promotes overfitting and poor-performing user guidelines. To address this shortcoming, we propose in this work a benchmark suite, OptimSuite, which covers a broad range of black-box optimization problems, ranging from academic benchmarks to real-world applications, from discrete over numerical to mixed-integer problems, from small to very large-scale problems, from noisy over dynamic to static problems, etc. We demonstrate the advantages of such a broad collection by deriving from it Automated Black Box Optimizer (ABBO), a general-purpose algorithm selection wizard. Using three different types of algorithm selection techniques, ABBO achieves competitive performance on all benchmark suites. It significantly outperforms previous state of the art on some of them, including YABBOB and LSGO. ABBO relies on many high-quality base components. Its excellent performance is obtained without any task-specific parametrization. The benchmark collection, the ABBO wizard, its base solvers, as well as all experimental data are reproducible and open source in OptimSuite.

## 1 Introduction: State of the Art

Many real-world optimization challenges are black-box problems; i.e., instead of having an explicit problem formulation, they can only be accessed through the evaluation of solution candidates. These evaluations often require simulations or even physical experiments. Black-box optimization methods are particularly widespread in machine learning (Salimans et al., 2016; Wang et al., 2020), to the point that it is considered a key research area of artificial intelligence. Black-box optimization algorithms are typically easy to implement and easy to adjust to different problem types. To achieve peak performance, however, proper algorithm selection and configuration are key, since black-box optimization algorithms have complementary strengths and weaknesses (Rice, 1976; Smith-Miles, 2009; Kotthoff, 2014; Bischl et al., 2016; Kerschke & Trautmann, 2018; Kerschke et al., 2018). But whereas automated algorithm selection has become standard in SAT solving (Xu et al., 2008) and AI planning (Vallati et al., 2015), a manual selection and configuration of the algorithms is still predominant in the broader black-box optimization context. To reduce the bias inherent to such manual choices, and to support the automation of algorithm selection and configuration, sound comparisons of the different black-box optimization approaches are needed. Existing benchmarking suites, however, are rather selective in the problems they cover. This leads to specialized algorithm frameworks whose performance suffer from poor generalizability. Addressing this flaw in black-box optimization, we present a unified benchmark collection which covers a previously unseen breadth of problem instances. We use this collection to develop a high-performing algorithm selection wizard, ABBO. ABBO uses high-level problem characteristics to select one or several algorithms, which are run for the allocated budget of function evaluations. Originally derived from a subset of the available benchmark collection, in particular YABBOB, the excellent performance of ABBO generalizes across almost all settings of our broad benchmark suite. Implemented as a fork of Nevergrad (Rapin & Teytaud, 2018), the benchmark collection, the ABBO wizard, the base solvers, and all performance data are open source. The algorithms are automatically rerun at certain time intervals and all

**Algorithm 1** High-level overview of ABBO. Selection rules are followed in this order, first match applied. $d$ = dimension, budget $b$ = number of evaluations. Details in (Anonymous, 2020).

| Case | Choice |
|---|---|
| **Discrete decision variables only** | |
| Noisy optimization with categorical variables | Genetic algorithm mixed with bandits (Heidrich-Meisner & Igel, 2009; Liu et al., 2020). |
| alphabets of size $< 5$, sequential evaluations | $(1 + 1)$-Evolutionary Alg. with linearly decreasing stepsize |
| alphabets of size $< 5$, parallel case | Adaptive $(1 + 1)$-Evolutionary Alg. (Doerr et al., 2019). |
| Other discrete cases with finite alphabets | Convert to the continuous case using SoftMax as in (Liu et al., 2020) and apply CMandAS2 (Rapin et al., 2019) |
| Presence of infinite discrete domains | FastGA (Doerr et al., 2017) |
| **Numerical decision variables only, evaluations are subject to noise** | |
| $d > 100$ | progressive optimization as in (Berthier, 2016). |
| $d \leq 30$ | TBPSA (Hellwig & Beyer, 2016) |
| $b > 100$ | sequential quadratic programming |
| Other cases | TBPSA (Hellwig & Beyer, 2016) |
| **Numerical decision variables only, high degree of parallelism** | |
| Parallelism $> b/2$ or $b < d$ | MetaTuneRecentering (Meunier et al., 2020) |
| Parallelism $> b/5$, $d < 5$, and $b < 100$ | DiagonalCMA-ES (Ros & Hansen, 2008) |
| Parallelism $> b/5$, $d < 5$, and $b < 500$ | Chaining of DiagonalCMA-ES (100 asks), then CMA-ES+meta-model (Auger et al., 2005) |
| Parallelism $> b/5$, other cases | NaiveTBPSA as in (Cauwet & Teytaud, 2020) |
| **Numerical decision variables only, sequential evaluations** | |
| $b > 6000$ and $d > 7$ | Chaining of CMA-ES and Powell, half budget each. |
| $b < 30d$ and $d > 30$ | $(1 + 1)$-Evol. Strategy w/ 1/5-th rule (Rechenberg, 1973) |
| $d < 5$ and $b < 30d$ | CMA-ES + meta-model (Auger et al., 2005) |
| $b < 30d$ | Cobyla (Powell, 1994) |
| **For all other cases and all details, please refer to the source code** | |

data is exported to the public dashboard (Rapin & Teytaud, 2020). For ICLR reviewers, all code is available, thanks to github-anonymizer, at (Anonymous, 2020).

In summary, our contributions are as follows. **(1) OptimSuite Benchmark Collection:** Optim-Suite combines several contributions that recently led to improved reliability and generalizability of black-box optimization benchmarking, among them LSGO (Li et al., 2013), YABBOB (Hansen et al., 2009; Liu et al., 2020; Anonymous, 2020), Pyomo (Hart et al., 2017; Anonymous, 2020), MLDA (Gallagher & Saleem, 2018), and MuJoCo (Todorov et al., 2012; Mania et al., 2018), and others (novelty discussed in Section 2). **(2) Algorithm Selection Wizard ABBO:** Our algorithm selection technique, ABBO, can be seen as an extension of the Shiwa wizard presented in (Liu et al., 2020). It uses three types of selection techniques: *passive algorithm selection* (choosing an algorithm as a function of a priori available features (Baskiotis & Sebag, 2004; Liu et al., 2020)), *active algorithm selection* (a bet-and-run strategy which runs several algorithms for some time and stops all but the strongest (Mersmann et al., 2011; Pitzer & Affenzeller, 2012; Fischetti & Monaci, 2014; Malan & Engelbrecht, 2013; Muñoz Acosta et al., 2015; Cauwet et al., 2016; Kerschke et al., 2018)), and *chaining* (running several algorithms in turn, in an a priori defined order (Molina et al., 2009)). Our wizard combines, among others, algorithms suggested in (Virtanen et al., 2019; Hansen & Ostermeier, 2003; Storn & Price, 1997; Powell, 1964; 1994; Liu et al., 2020; Hellwig & Beyer, 2016; Artelys, 2015; Doerr et al., 2017; 2019; Dang & Lehre, 2016). Another core contribution of our work is a sound comparison of our wizard to Shiwa, and to the long list of algorithms available in Nevergrad.

## 2 SOUND BLACK-BOX OPTIMIZATION BENCHMARKING

We summarize desirable features and common shortcomings of black-box optimization benchmarks and discuss how OptimSuite addresses these.

**Generalization.** The most obvious issue in terms of generalization is the statistical one: we need sufficiently many experiments for conducting valid statistical tests and for evaluating the robustness of algorithms' performance. This, however, is probably not the main issue. A biased benchmark, excluding large parts of the industrial needs, leads to biased conclusions, no matter how many experiments we perform. Inspired by (Recht et al., 2018) in the case of image classification, and similar to the spirit of cross-validation for supervised learning, we use a much broader collection of benchmark problems for evaluating algorithms in an unbiased manner. Another subtle issue in terms of generalization is the case of instance-based choices of (hyper-)parameters: an experimenter

| Testbed | BBOB | MuJoCo | LSGO | Nevergrad | BBComp | OptimSuite |
|---|---|---|---|---|---|---|
| Large scale | - | | + | + | | + |
| Translations | + | | + | + | + | + |
| Symmetrizations / rotations | + | | + | + | | + |
| One-line reproducibility | - | | - | + | | + |
| Periodic automated dashboard | | | | + | | + |
| Complex or real-world | - | + | - | + | | + |
| Multimodal | + | + | + | + | + | + |
| Open sourced / no license | | - | | | | + |
| Ask/tell/recommend correct | - | | + | + | + | + |
| Far-optimum | + | | | + | | + |
| Human excluded / client-server | | | | | + | |

Table 1: Properties of selected benchmark collections (details in appendix A). "+" means that the feature is present, "-" that the feature is missing, and an empty case that it is not applicable. Far-optimum refers to problems with optimum far from the center or on the side of the domain; such benchmarks test the ability of optimization algorithms to answer promptly to a bad initialization (Chotard et al., 2012). "Translations" applies only to artificial benchmarks. A "+" in rows "Multimodal", "symmetrization", and "real-world" does not imply that *all* test functions have this property. "Open sourced" refers to open access to most algorithms involved in the published comparison; here, "-" refers to license issues for the benchmark itself.

modifying the algorithm or its parameters specifically for each instance can easily improve results by a vast margin. In this paper, we consider that only the following problem properties are known in advance (and can hence be used for algorithm selection and configuration): the dimension of the domain, the type and range of each variable, their order, the presence of noise (but not its intensity), the budget, the degree of parallelism (i.e., number of solution candidates that can be evaluated simultaneously). To mitigate the common risk of over-tuning, we evaluate algorithms on a broad range of problems, from academic benchmark problems to real-world applications. Each algorithm runs on all benchmarks without any change or task-specific tuning.

**Use the ask, tell, and recommend pattern.** Formalizing the concept of numerical optimization is typically made through the formalism of oracles or parallel oracles (Rogers, 1987). A recent trend is the adoption of the ask-and-tell format (Collette et al., 2010). The bandit literature pointed out that we should distinguish *ask*, *tell*, and *recommend*: the way we choose a point for gathering more information is not necessarily close to the way we choose an approximation of the optimum (Bubeck et al., 2011; Coulom, 2012b; Decock & Teytaud, 2013). We adopt the following framework: given an objective function $f$ and an *optimizer*, for $i \in \{1, \dots, T\}$, do $x \leftarrow optimizer.ask$ and $optimizer.tell(x, f(x))$. Then, evaluate the performance with $f(optimizer.recommend)$. The requirement of a recommend method distinct from the ask is critical in noisy optimization. A debate pointed out some shortcomings in the the noisy counterpart of BBOB (Auger & Hansen, 2009) which was assuming that $ask = recommend$: (Beyer, 2012a;b; Coulom, 2012a) have shown that in the noisy case, this difference was particularly critical, and a framework should allow algorithms to "recommend" differently than they "ask". A related issue is that a run with budget $T$ is not necessarily close to the truncation of a run in budget $10T$.

**Translation-invariance.** Zero frequently plays a special role in optimization. For example, complexity penalizations often "push" towards zero. In control, numbers far from zero are often more likely to lead to bang-bang solutions (and therefore extract zero signal, leading to a needle-in-the-haystack optimization situation), in particular with neural networks. In one-shot optimization, (Cauwet et al., 2019; Meunier et al., 2020) have shown how much focusing to the center is a good idea in particular in high-dimension. Our experiments in control confirm that the scale of the optimization search is critical, and explains the misleading results observed in some optimization papers (Section 4.2). In artificial experiments, several classical test functions have their optimum in zero. To avoid misleading conclusions, it is now a standard procedure, advocated in particular in (Hansen et al., 2009), to randomly translate the objective functions. This is unfortunately not always applied.

**Rotation and symmetrization.** Some optimization methods may perform well on separable objective functions but degrade significantly in optimizing non-separable functions. If the dimension of a separable objective function is $d$, these methods can reduce the objective function into $d$ one-dimensional optimization processes (Salomon, 1996). Therefore, Hansen et al. (2009; 2011) have

insisted that objective functions should be rotated to generate more difficult non-separable objective functions. However, Bousquet et al. (2017) pointed out the importance of dummy variables, which are not invariant per rotation; and (Holland, 1975) and more generally the genetic algorithms literature insist that rotation does not always makes sense – we lose some properties of a real-world objective function, and in some real-world scenarios rotating would, e.g., mix temperature, distance and electric intensity. Permutating the order of variables is also risky, as their order is sometimes critical - $k$-point crossovers a la Holland (Holland, 1975) typically assume some order of variables, which would be broken. Also, users sometimes rank variables with the most important first – and some optimization methods do take care of this (Cauwet et al., 2019). In OptimSuite, we do include rotations, but include both cases, rotated or not. For composite functions which use various objective functions on various subsets of variables, we consider the case with rotations – without excluding the non-rotated case. An extension of symmetrization that we will integrate later in ABBO, which makes sense for replicating an objective function without exact identity, consists in symmetrizing some variables: for example, if the $i^{th}$ variable has range $[a, b]$, we can replace $x_i$ by $b + a - x_i$. Applying this on various subsets of variables leads to $2^d$ symmetries of an objective function, if the dimension is $d$. This variation can reduce the bias toward symmetric search operations (Li et al., 2013).

**Benchmarking in OptimSuite.** We summarize in Table 1 some existing benchmark collections and their (desirable) properties. We inherit various advantages from Nevergrad, namely the automatic rerun of experiments and reproducibility in one-line. Our fork includes PBT (a small scale version of Population-Based Training (Jaderberg et al., 2017)), Pyomo (Hart et al., 2017), Photonics (problems related to optical properties and nanometric materials), YABBOB and variants, LSGO (Li et al., 2013), MLDA (Gallagher & Saleem, 2018), PowerSystems, FastGames, 007, Rocket, SimpleTSP, Realworld (Liu et al., 2020), MuJoCo (Todorov et al., 2012) and others including a (currently small) benchmark of hyperparameters of Scikit-Learn (Pedregosa et al., 2011) and Keras-tuning, all of those being visible for review at the above-mentioned anonymized URL (underlined means: the benchmark is either new, or, in the case of PowerSystems or SimpleTSP, significantly modified compared to previous works, or, in the case of LSGO or MuJoCo, included for the first time inside Nevergrad. For MuJoCo, we believe that interfacing with Nevergrad is particularly useful, to ensure fair comparisons, which rely very much on the precise setup of MuJoCo. . We note that, at present, we do not reproduce the extreme black-box nature of Loshchilov & Glasmachers (2017). Still, by integrating such a wide range of benchmarks in a single open source framework, which, in addition, is periodically re-run, we believe that Nevergrad/OptimSuite provides a significant contribution to benchmarking, and this both for the optimization and the machine learning community, where most of the benchmark suites originate from.

## 3    A New Algorithm Selection Wizard: ABBO

Black-box optimization is sometimes dominated by evolutionary computation. Evolution strategies (Beyer & Schwefel, 2002; Beyer, 2001; Rechenberg, 1973) have been particularly dominant in the continuous case, in experimental comparisons based on the Black-Box Optimization Benchmark BBOB (Hansen et al., 2009) or variants thereof. Parallelization advantages (Salimans et al., 2016) are particularly appreciated in the machine learning context. However, Differential Evolution (Storn & Price, 1997) is a key component of most winning algorithms in competitions based on variants of Large Scale Global Optimization (LSGO (Li et al., 2013)), suggesting a significant difference between these benchmarks. In particular, LSGO is more based on correctly identifying a partial decomposition and scaling to $\geq$ 1000 variables, whereas BBOB focuses (mostly, except (Varelas et al., 2018)) on $\leq$ 40 variables. Mathematical programming techniques (Powell, 1964; 1994; Nelder & Mead, 1965; Artelys, 2015) are rarely used in the evolutionary computation world, but they have won competitions (Artelys, 2015) and significantly improved evolution strategies through memetic methods (Radcliffe & Surry, 1994). Algorithm selection was applied to continuous black-box optimization and pushed in Nevergrad Liu et al. (2020) : their optimization algorithm combines many optimization methods and outperforms each of them when averaged over diverse test functions. Closer to machine learning, efficient global optimization (Jones et al., 1998) is widely used, although it suffers from the curse of dimensionality more than other methods (Snoek et al., 2012); (Wang et al., 2020) presented a state-of-the-art algorithm in black-box optimization on MuJoCo, i.e., for the control of various realistic robots (Todorov et al., 2012). We propose ABBO, which extends (Liu et al., 2020) by the following features: (1) Better use of chaining (Molina et al.,

| Benchmark | Use for ABBO | # of configs | ranking | | | ABBO best competitor |
|---|---|---|---|---|---|---|
| | | | ABBO | Shiwa | CMA-ES | |
| HDBO | Designing | 24 | 2/21 | $1^\dagger$ | 2 | Shiwa |
| PARAMULTIMODAL | Designing | 112 | **1/27** | $3^\dagger$ | 6 | DiagonalCMA-ES (Ros & Hansen, 2008) |
| Realworld | Designing | 486 | **1/6** | $2^\dagger$ | 3 | Shiwa (Liu et al., 2020) |
| Illcondi | Designing | 12 | **1/24** | $3^\dagger$ | 8 | Cobyla |
| Illcondipara | Designing | 12 | 5/28 | $7^\dagger$ | 3 | DiagonalCMA-ES |
| YABBOB | Designing* | 630 | **1/8** | 2 | 5 | Shiwa |
| YAPARABBOB | Designing* | 630 | **1/6** | 4 | 5 | MetaModel |
| YAHDBBOB | Designing* | 378 | 2/19 | 3 | 18 | $(1+1)$-ES |
| YANOISYBBOB | Designing* | 630 | 2/11 | 6 | 10 | SQP |
| YAHDNOISYBBOB | Designing* | 630 | 4/24 | 2 | 13 | SQP |
| YASMALLBBOB | Designing* | 378 | **1/8** | 2 | 7 | Shiwa |
| HdMultimodal | Validation | 42 | **1/14** | $2^\dagger$ | 4 | Shiwa |
| Noisy | Validation | 96 | 16/28 | $19^\dagger$ | NA | RecombiningOptimisticNoisyDiscrete$(1+1)$ |
| RankNoisy | Validation | 72 | 4/25 | NA | 19 | ProgD13 |
| AllDEs | Validation | 60 | **1/28** | $2^\dagger$ | 3 | Shiwa (Liu et al., 2020) |
| Pyomo | Evaluating | 104 | **1/19** | $3^\dagger$ | 10 | Shiwa (Liu et al., 2020) |
| Rocket | Evaluating | 13 | 5/18 | $4^\dagger$ | 3 | DiagonalCMA-ES (Ros & Hansen, 2008) |
| SimpleTSP | Evaluating | 52 | 3/15 | $2^\dagger$ | 7 | PortfolioDiscrete$(1+1)$ |
| Seq. Fastgames | Evaluating | 20 | 3/28 | $4^\dagger$ | 23 | OptimisticDiscrete$(1+1)$ |
| LSGO | Evaluating | 45 | **1/6** | $4^\dagger$ | 6 | MiniLHSDE |
| Powersystems | Evaluating | 48 | 10/26 | NA | 25 | $(1+1)$-ES |

Table 2: Nevergrad maintains a dashboard (Rapin & Teytaud, 2020). For each experiment, there are many configurations (budget, objective function, possibly dimension, and noise level). We separate benchmarks used for designing ABBO, benchmarks used for validation, and those only used for testing. "*" denotes benchmarks used for designing Shiwa (which is used inside ABBO). We present the rank based on the winning rate of ABBO in the dashboard. Since the submission of this paper, several variants of bandit-based algorithms have been added for high-dimensional noisy optimization. They outperform ABBO, hence its poor rank for these cases. Detailed plots are available in Appendix B. As expected, DE variants are strong on LSGO and CMA-ES variants are strong for YABBOB. ABBO also performs well on YABBOB, which was used for designing its ancestor Shiwa (see Fig. 1). For the MuJoCo testbed, details are available in Table 3 and Figure 3. Our modifications in the codebase implies an improvement of Shiwa compared to the version published in (Liu et al., 2020); for example, our chaining implies that the $(k+1)^{th}$ code starts from the best point obtained by the $k^{th}$ algorithm, which significantly improves in particular the chaining CMA-ES+Powell or CMA-ES+SQP. Experiments with "$\dagger$" in the ranking of Shiwa correspond to this improved version of Shiwa.

2009) and more intensive use of mathematical programming techniques for the last part of the optimization run, i.e., the local convergence, thanks to Meta-Models (in the parallel case) and more time spent in Powell's method (in the sequential case). This explains the improvement visible in Section 4.1. (2) Better performance in discrete optimization, using additional codes recently introduced in Nevergrad, in particular adaptive step-sizes. (3) Better segmentation of the different cases of continuous optimization. We still entirely rely on the base algorithms as available in Nevergrad; that is, we did not modify the tuning of any method. We acknowledge that our method only work thanks to the solid base components available in Nevergrad, which are based on contributions from various research teams. The obtained algorithm selection wizard, ABBO, is presented in Algorithm 1. The performances of ABBO is summarized in Table 2 and a detailed dashboard is available at https://dl.fbaipublicfiles.com/nevergrad/allxps/list.html.

## 4 EXPERIMENTAL RESULTS

When presenting results on a single benchmark function, we present the usual average objective function value for different budget values. When a collection comprises multiple benchmark problems, we present our aggregated experimental results with two distinct types of plots: (1) Average normalized objective value for each budget, averaged over all problems. The normalized objective value is the objective value linearly rescaled to $[0, 1]$. (2) Heatmaps, showing for each pair (x, y) of optimization algorithms the frequency at which Algorithm x outperforms Algorithm y. Algorithms are ranked by average winning frequency. We use red arrows to highlight ABBO.

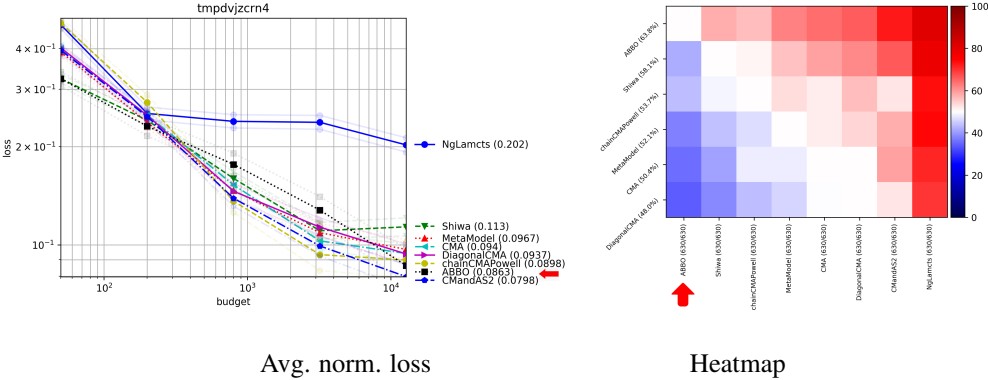

Avg. norm. loss                    Heatmap

Figure 1: Average normalized loss and heatmap for YABBOB. Additional plots for High-dimensional (HD), NoisyHD, and Large budgets are available in Appendix B. Other variants include parallel, differences of budgets, and combinations of those variants, with excellent results for ABBO (see Nevergrad's dashboard `https://dl.fbaipublicfiles.com/nevergrad/allxps/list.html`, publicly visible with anonymized entries, for the part of our code which is already merged).

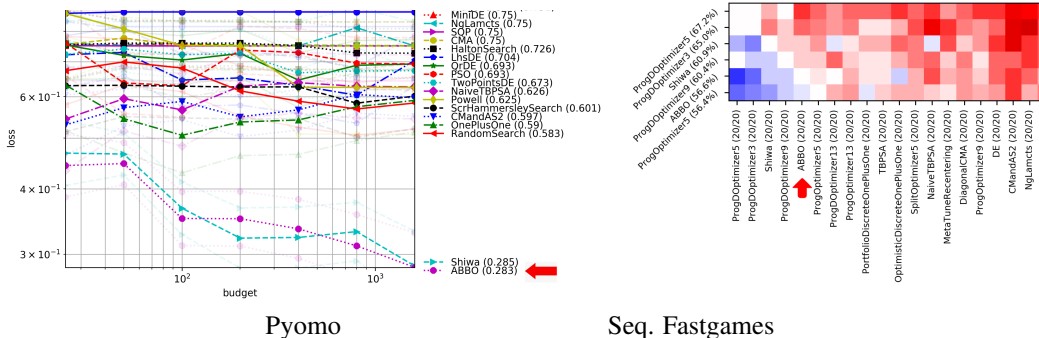

Pyomo                        Seq. Fastgames

Figure 2: Additional problems: Pyomo (covering Knapsack, P-median and others), Sequential-Fastgames (presented as heatmaps due to the high noise: GuessWho, War, Batawaf, Flip). Rockets, SimpleTSP, PowerSystems, and LSGO plots are available in Appendix B, Figures 7, and 8. Pyomo and SimpleTSP include discrete variables. Pyomo includes constraints. Rocket, PowerSystems, SequentialFastGames are based on open source simulators and are already merged from OptimSuite to Nevergrad.

### 4.1 BENCHMARKS IN OPTIMSUITE USED FOR DESIGNING AND VALIDATING ABBO

**YABBOB** (Yet Another Black-Box Optimization Benchmark (Rapin et al., 2019)), is an adaptation of BBOB (Hansen et al., 2009), with extensions such as parallelism and noise management. It contains many variants, including noise, parallelism, high-dimension (BBOB was limited to dimension $< 50$). Several extensions, for the high-dimensional, the parallel or the big budget case, have been developed: we present results in Figures 1 and 4. The high-dimensional one is inspired by (Li et al., 2013), the noisy one is related to the noisy counterpart of BBOB but correctly implements the difference between ask and recommend as discussed in Section 2. The parallel one generalizes YABBOB to settings in which several evaluations can be executed in parallel. Results on PARA-MULTIMODAL are presented in Figure 6 (left). In addition, ABBO was run on ILLCONDI & ILLCONDIPARA (ill conditionned functions), HDMULTIMODAL (a multimodal case focusing on high-dimension), NOISY & RANKNOISY (two noisy continuous testbeds), YAWIDEBBOB (a broad range of functions including discrete cases and cases with constraints).

| Task | Target | LA-MCTS results | ABBO result | LA-MCTS avg reward | ABBO avg reward |
|------|--------|-----------------|-------------|--------------------|-----------------|
| Swimmer-v2 | 325 | 132 ioa | around 450 iter | 358 | **365** |
| Hopper-v2 | 3120 | 2897 ioa | around 3000 iter | **3292** | 1787 |
| HalfCheetah-v2 | 3430 | 3877 ioa | around 4000 iter | 3227 | **4730** |
| Walker2d-v2* | 4390 | BR: 3314 (not reached) | BR: **4398, budget** $< 64000$ **(reached!)** | 2769 | **2949** |
| Ant-v2* | 3580 | BR: 2791 (not reached) | BR: **5325, budget** $< 32000$ **(reached!)** | 2511 | **3532** |
| Humanoid-v2* | 6000 | BR: 3384 (not reached) | BR (budget 500000): **4870** | 2511 | **4620** |

Table 3: Results for a linear policy in the black-box setting from the latest black-box paper (Wang et al., 2020) and references therein, compared to results from ABBO. Two last columns = average reward for the maximum budget tested in (Wang et al., 2020), namely 1k, 4k, 4k, 40k, 30k, 40k, respectively. "ioa" = iterations on average for reaching the target. "iter" = iterations for target reached for median run. "*" refers to problems for which the target was not reached by Wang et al. (2020): then BR means "best result in 10 runs". ABBO reaches the target for Humanoid and Ant whereas previous (black-box) papers did not; we get nearly the same ioa for Hopper and HalfCheetah (Nevergrad computed the expected value instead of computing the ioa, so we cannot compare exactly; see Figure 3 for curves). ABBO is slower than LA-MCTS on Swimmer. Note that we keep the same method for all benchmarks whereas LA-MCTS modified the algorithm for 3 rows. On HDMULTIMODAL, ABBO performs better than LA-MCTS, as detailed in the text, and as confirmed in (Wang et al., 2020), which acknowledges the poor results of LA-MCTS for high-dimensional Ackley and Rosenbrock.

**AllDEs and Hdbo** are benchmark collections specifically designed to compare DE variants (AllDEs) and high-dimensional Bayesian Optimization (Hdbo), respectively (Rapin & Teytaud, 2018). These benchmark functions are similar to the ones used in YABBOB. Many variants of DE (resp. BO) are considered. Results are presented in Figure 5. They show that the performance of ABBO, relatively to DE or BO, is consistent over a wide range of parametrizations of DE or BO, at least in their most classical variants. All these variants are publicly visible in Nevergrad and/or in our anonymized branch.

**Realworld:** A test of ABBO is performed on the Realworld optimization benchmark suite proposed in (Rapin & Teytaud, 2018). This suite includes testbeds from MLDA (Gallagher & Saleem, 2018) and from (Liu et al., 2020). Results for this suite, presented in Figure 6, confirm that ABBO performs well also on benchmarks that were not explicitly used for its design - however, this benchmark was used for designing Shiwa, which was the basis of our ABBO. A rigorous cross-validation, on benchmarks totally independent from the design of Shiwa, is provided in the next sections.

## 4.2 NEW BENCHMARKS IN OPTIMSUITE USED ONLY FOR EVALUATING ABBO

**Pyomo** is a modeling language in Python for optimization problems (Hart et al., 2017). It is popular and has been adopted in formulating large models for complex and real-world systems, including energy systems and network resource systems. We implemented an interface to Pyomo for Nevergrad and enriched our benchmark problems (Anonymous, 2020), which include discrete variables and constraints. Experimental results are shown in Figure 2. They show that ABBO also performs decently in discrete settings and in constrained cases.

**Additional new artificial and real-world functions:** LSGO (large scale global optimization) combines various functions into an aggregated difficult testbed including composite highly multimodal functions. Correctly decomposing the problem is essential. Various implementations of LSGO exist; in particular we believe that some of them do not match exactly. Our implementation follows (Li et al., 2013), which introduces functions with subcomponents (i.e., groups of decision variables) having non-uniform sizes and non-uniform, even conflicting, contributions to the objective function. Furthermore, we present here experimental results on SequentialFastgames from the Nevergrad benchmarks, and three newly introduced benchmarks, namely Rocket, SimpleTSP (a set of traveling salesman problems), power systems (unit commitment problems (Padhy, 2004)). Experimental results are presented in Figures 2, 7, and 8. They show that ABBO performs well on new benchmarks, never used for its design nor for that of the low-level heuristics used inside ABBO.

**MuJoCo.** Many articles (Sener & Koltun, 2020; Wang et al., 2020) studied the MuJoCo testbeds (Todorov et al., 2012) in the black-box setting. MuJoCo tasks correspond to control problems. Defined in (Wang et al., 2020; Mania et al., 2018), the objective is to learn a linear mapping

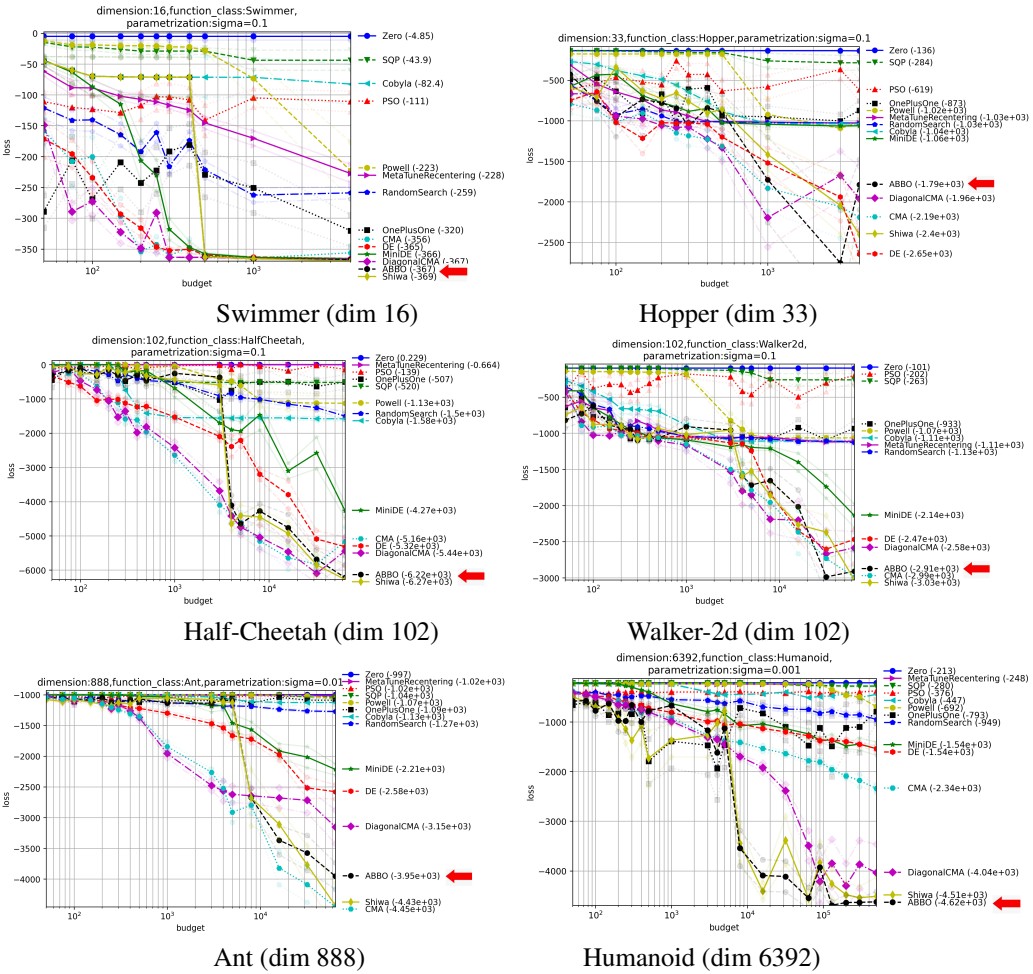

Figure 3: Results on the MuJoCo testbeds. Dashed lines show the standard deviation. Compared to the state of the art in (Wang et al., 2020), with an algorithm adapted manually for the different tasks, we get overall better results for Humanoid, Ant, Walker. We get worse results for Swimmer (could match if we had modified our code for the 3 easier tasks as done in (Wang et al., 2020)), similar for Hopper and Cheetah: we reach the target for 5 of the 6 problems (see text). Runs of Shiwa correspond to the improvement of Shiwa due to chaining, as explained in Table 2.

from states to actions. It turned out that the scaling is critical (Mania et al., 2018): for reasons mentioned in Section 2, solutions are close to 0. We chose to scale all the variables of the problem at the power of $0.1$ the closest to $1.2/d$, for all methods run in Figure 3. We remark that ABBO and Shiwa perform well, including comparatively to gradient-based methods in some cases, while having the ability to work when the gradient is not available. When the gradient is available, black-box methods do not require computation of the gradient, which saves time.

We use the same experimental setup as Wang et al. (2020) (linear policy, offline whitening of states). We get results better than LA-MCTS, in a setting i.e., does not use any expensive surrogate model (Table 3). Our runs with CMA-ES and Shiwa are better than those in (Wang et al., 2020). We acknowledge that LMRS (Sener & Koltun, 2020) outperforms our method on all MuJoCo tasks, using a deep network as a surrogate model: however, we point out that a part of their code is not open sourced, making the experiments not reproducible. In addition, when rerunning their repository without the non open sourced part, it solved Half-Cheetah within budget 56k, which is larger than ours. For Humanoid, the target was reached at 768k, which is again larger than our budget. Results from ABBO are comparable to, and usually better than (for the 3 hardest problems), results from LA-MCTS, while ABBO is entirely reproducible. In addition, it runs the same method for all

benchmarks and it is not optimized for each task specifically as in (Sener & Koltun, 2020; Wang et al., 2020). In contrast to ABBO, (Wang et al., 2020) uses different underlying regression methods and sampling methods depending on the MuJoCo task, and it is not run on other benchmarks except for some of the HDMULTIMODAL ones. On the latter, ABBO performances are significantly better for Ackley and Rosenbrock in dimension 100 (expected results around 100 and $10^{-8}$ after 10k iterations for Rosenbrock and Ackley respectively for ABBO, vs 0.5 and 500 in (Wang et al., 2020)). From the curves in (Wang et al., 2020) and in the present work, we expect LA-MCTS to perform well with an adapted choice of parametrization and with a low budget, for tasks related to MuJoCo, whereas ABBO is adapted for wide ranges of tasks and budgets.

## 5 CONCLUSIONS

This paper proposes OptimSuite, a very broad benchmark suite composed of real-world and of artificial benchmark problems. OptimSuite is implemented as a fork of Nevergrad (Rapin & Teytaud, 2018), from which it inherits a strong reproducibility: our (Python) code is open source (Anonymous, 2020), tests are rerun periodically, and up-to-date results are available in the public dashboard (Rapin & Teytaud, 2020). A whole experiment can be done as a one-liner. OptimSuite fixes several issues of existing benchmarking environments. The suite subsumes MuJoCo, Pyomo, LSGO, YABBOB, MLDA, and several new real-world problems. We also propose ABBO, an improved algorithm selection wizard. Despite its simplicity, ABBO shows very promising performance across the whole benchmark suite, often outperforming the previous state-of-the-art, problem-specific solvers: (a) by solving 5 of the 6 cases without any task-specific hyperparameter tuning, ABBO outperforms LA-MCTS, which was specialized for each single task, (b) ABBO outperforms Shiwa on YABBOB and its variants, which is the benchmark suite used to design Shiwa in the first place, (c) ABBO is also among the best methods on LSGO and almost all other benchmarks.

**Further work.** OptimSuite subsumes most of the desirable features outlined in Section 2, with one notable exception, the true black-box setting, which other benchmark environments have implemented through a client-server interaction (Loshchilov & Glasmachers, 2017). A possible combination between our platform and such a challenge, using the dashboard to publish the results, could be useful, to offer a meaningful way for cross-validation. Further improving ABBO is on the roadmap. In particular, we are experimenting with the automation of the still hand-crafted selection rules. Note, though, that it is important to us to maintain a high level of interpretability, which we consider key for a wide acceptance of the wizard. Another avenue for future work is a proper configuration of the low-level heuristics subsumed by ABBO. At present, some of them are merely textbook implementations, and significant room for improvement can therefore be expected. Newer variants (Loshchilov, 2014; Akimoto & Hansen, 2016; Loshchilov et al., 2018) of CMA-ES, of LMRS (Sener & Koltun, 2020), recent Bayesian optimization libraries (e.g. Eriksson et al. (2019)), as well as per-instance algorithm configuration such as Belkhir et al. (2017) are not unlikely to result in important improvements for various benchmarks. We also plan on extending OptimSuite further, both through interfacing existing benchmark collections/problems, and by designing new benchmark problems ourselves.

### ACKNOWLEDGMENTS

We thank the anonymous reviewers for valuable suggestions that helped us improve the clarity and the presentation of our work.

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

## A    DETAILS ABOUT PROPERTIES OF BENCHMARKS

We specify the properties mentioned in Table 1.

- **Large scale:** includes dimension $\geq 1000$.

- **Translations:** in unbounded continuous domains, a standard deviation $\sigma$ has to be provided, for example for sampling the first and second iterates of the optimization algorithm. Given a standard deviation $\sigma$, we consider that there is translation when optimas are randomly translated by a $\mathcal{N}(0, \sigma^2)$ shift. Only interesting for artificial cases.

- **Far-optimum:** optima are translated far from the optimum, with standard deviation at least $\mathcal{N}(0, 25 \times \sigma^2)$.

- **Symmetrizations / rotations (here assuming an optimum, up to translation, in** $0$**).** Rotation: with a random rotation matrix $M$, the function $x \mapsto f(x)$ is replaced by $x \mapsto f(M(x))$. Symmetrization: $x \mapsto f(x)$ can be replaced by $x \mapsto f(S(x))$, with $S$ a diagonal matrix with each diagonal coefficient equal to $1$ or $-1$ with probability $50\%$. We do not request all benchmarks to be rotated: it might be preferable to have both cases considered.

- **One-line reproducibility:** Where reproducibility requires significant coding, it is unlikely to be of great use outside of a very small set of specialists. One-line reproducibility is given when the effort to reproduce an entire experiment does not require more than the execution of a single line. We consider this to be an important feature.

- **Periodic automated dashboard:** are algorithms re-run periodically on new problem instances? Some platforms do not collect the algorithms, and reproducibility is hence not given. An automated dashboard is convenient also because new problems can be added "on the go" without causing problems, as all algorithms will be executed on all these new problem instances. This feature addresses what we consider to be one of the biggest bottlenecks in the current benchmarking environments.

- **Complex or real-world:** Real-world is self-explanatory; complex means a benchmark involving a complex simulator, even if it is not real world. MuJoCo is in the "complex" category.

- **Multimodal:** whether the suite contains problems for which there are local optima which are not global optima.

- **Open sourced / no license:** Are algorithms and benchmarks available under an open source agreement. BBOB does not collect algorithms, MuJoCo requires a license, LSGO and BBOB are not realworld, Mujoco requires a license, BBComp is no longer maintained, Nevergrad before OptimSuite did not include complex ML problems without license issue before our work: some people have already applied Nevergrad to MuJoCo, but with our work MuJoCo becomes part of Nevergrad so that people can upload their code in Nevergrad and it will be run on all benchmarks, including MuJoCo.

- **Ask/tell/recommend correctly implemented** (Collette et al., 2010; Bubeck et al., 2011): The ask and tell idea (developed in Collette et al. (2010)) is that an optimization algorithm should not come under the format $Optimizer.minimize(objective - function)$ because there are many settings in which this is not possible: you might think of agents optimizing concurrently their own part of an objective function, and problems of reentrance, or asynchronicity. All settings can be recovered from an ask/tell optimization method. This becomes widely used. However, as well known in the bandit literature (you can think of pure exploration bandits (Bubeck et al., 2011)), it is necessary to distinguish ask, tell and recommend: the "recommend" method is the one which proposes an approximation of the optimum. Let us develop an example explaining why this matters: the domain is $\{1, 2, 3, 4\}$, and we have a budget of $20$ in a noisy case. NoisyBBOB assumes that the optimum is found when "ask" returns the optimum arm: then, the status remains "found" even if the algorithm has no idea where is the optimum and never comes back nearby. So an algorithm which just iteratively "asks" $1, 2, 3, 4, 1, 2, 3, 4, \ldots$ reaches the optimum in at most $4$ iterations. This does not mean anything in the noisy case, as the challenge is to figure out which of the four numbers is the optimum. With a proper ask/tell/recommend, the optimizer chooses an arm at the end of the budget. A simple regret is then computed.

Actually this also matters in the noise-free case, but the issue is much more critical in noisy optimization. The case of continuous noisy optimization also has counter-examples and all the best noisy optimization algorithms use ask/tell/recommend. We add the reference to the paper above.

- **Human excluded / client-server:** The problem instances are truly black-box. Algorithms can only suggest points and observe function values, but neither the algorithm nor its designer have access to any other information about the problem apart from the number of variables, their type, ranges, and order. It is impossible to repeat experiments for tuning hyperparameters without "paying" the budget of the HP tuning. This is something we could not do, as everything is public and open sourced: however, we believe that we mitigate this issue by considering a large number of benchmarks.

## B ADDITIONAL FIGURES

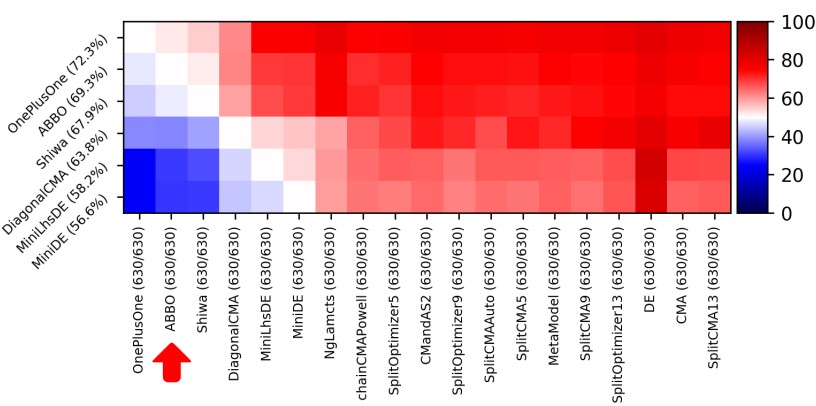

High-dimensional (HD)

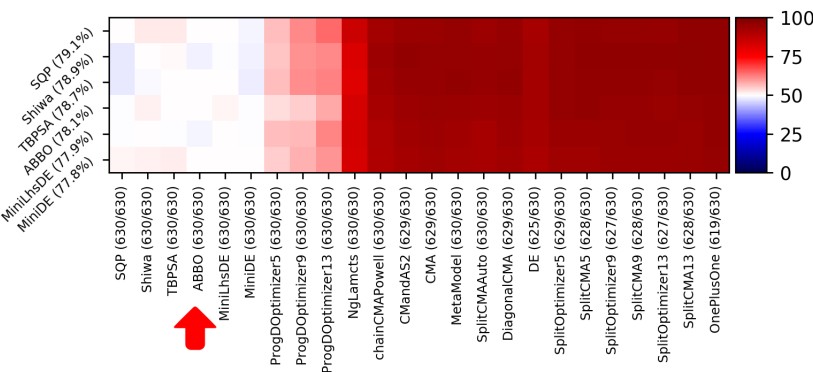

Noisy HD

Figure 4: YAHDBBBOB (dimension $\geq 50$) and YANOISYHDBBBOB (noisy + dimension $\geq 50$) heatmaps.

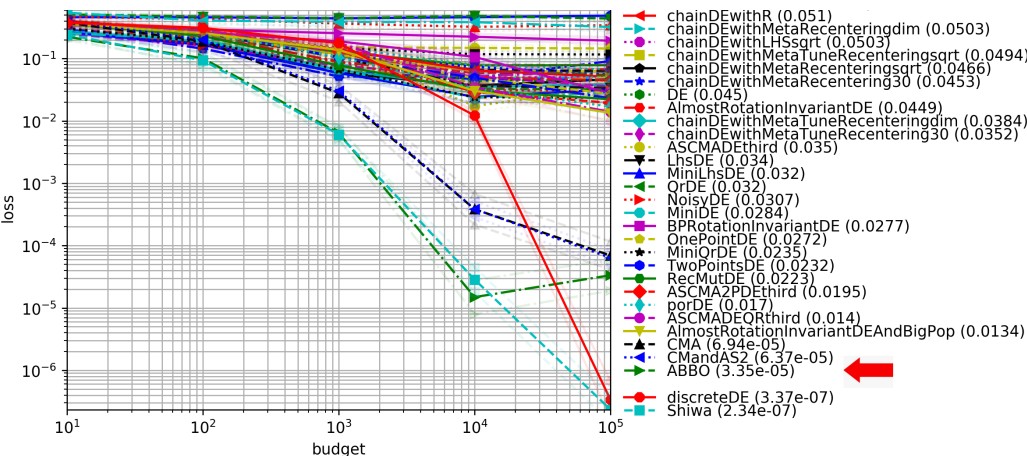

AllDEs ($d \in \{5, 20, 100\}$.)

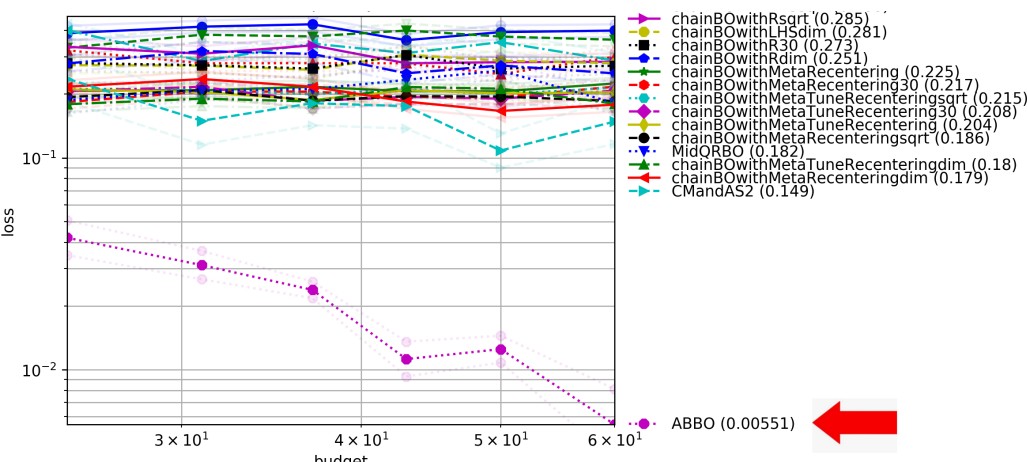

HDBO ($d \in \{20, 2000\}$)

Figure 5: ABBO vs specific families of optimization algorithms (DE, and BO in the high-dimensional case) on Cigar, Hm, Ellipsoid, Sphere functions. Not all run algorithms are mentioned, for short. Bayesian optimization (Nevergrad uses Nogueira (2014–)), often exploring boundaries first, is outperformed in high dimension (Wang et al., 2020).

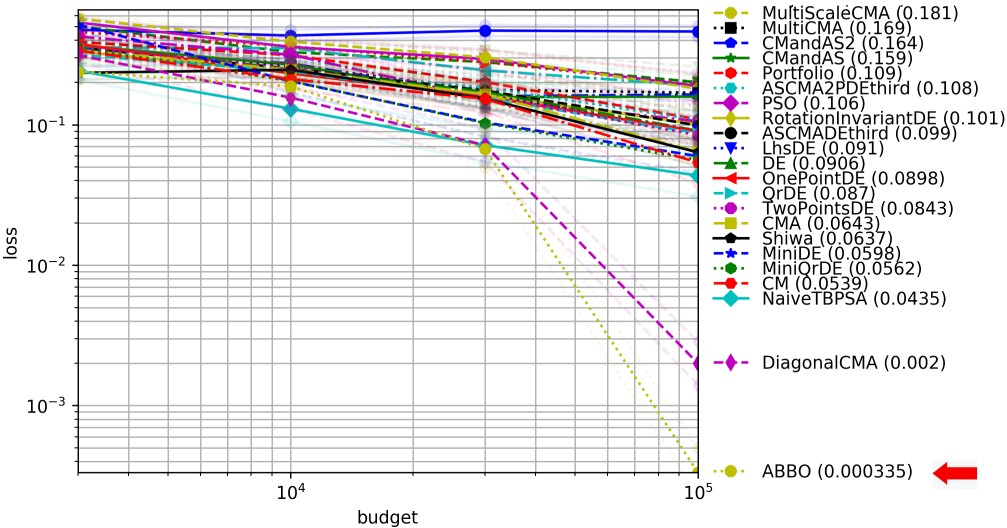

PARAMULTIMODAL

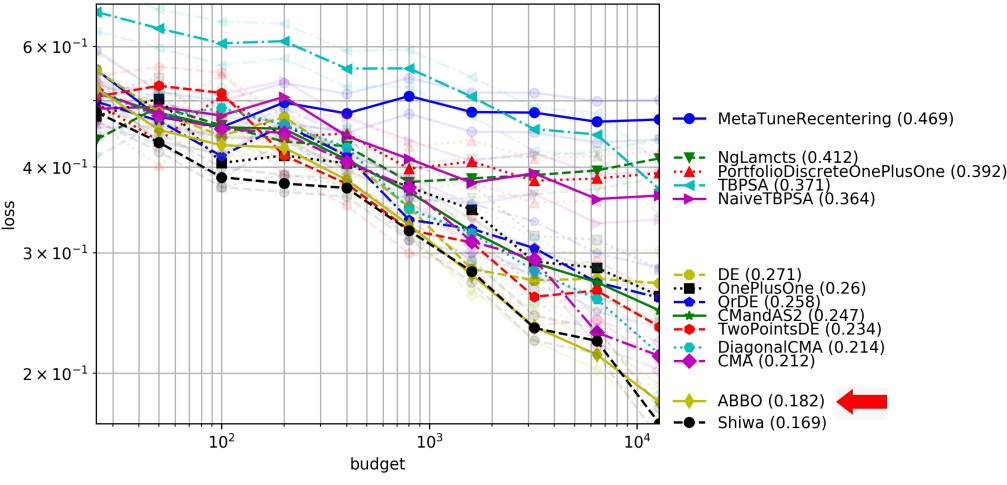

Realworld

Figure 6: Up: experiments for the parallel multimodal setting PARAMULTIMODAL. Budget up to 100000, parallelism 1000, Ackley+Rosenbrock+DeceptiveMultimodal+Griewank+Lunacek+Hm. Bottom: Realworld benchmark from Nevergrad: games, Sammon mappings, clustering, small traveling salesman instance, small power systems.

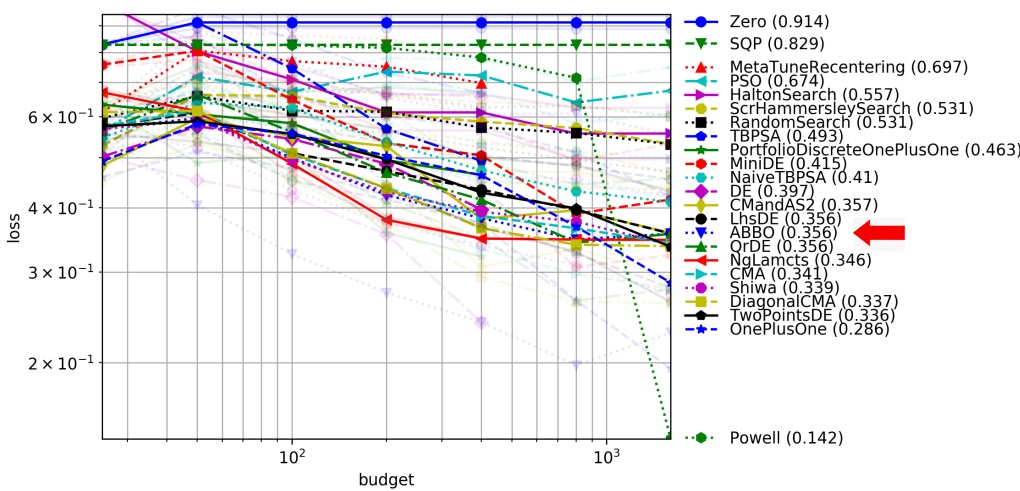

Rocket

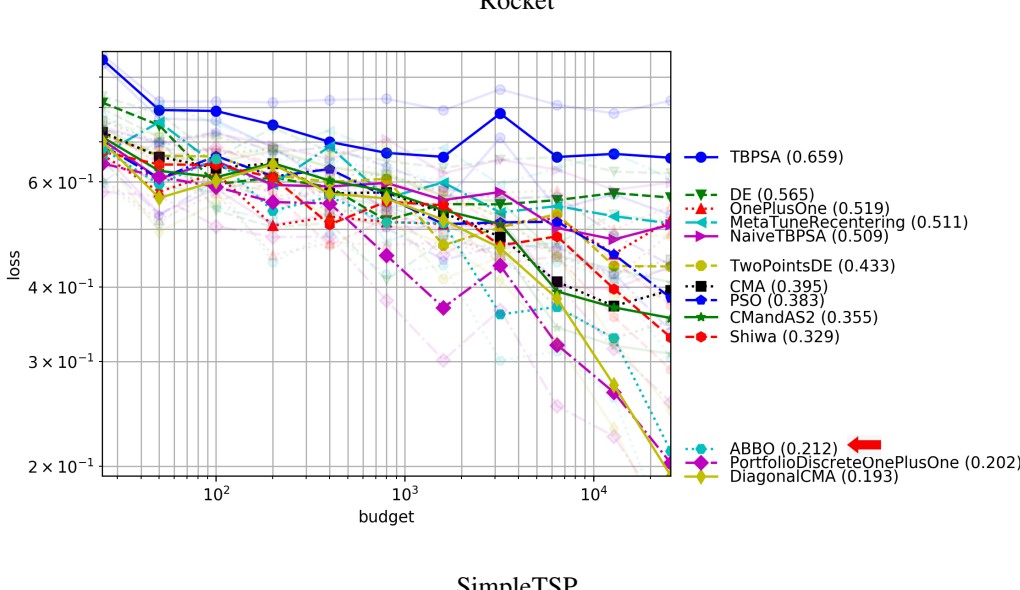

SimpleTSP

Figure 7: Additional problems (1): Rocket (26 continuous variables, budget up to 1600, sequential or parallelism 30) and SimpleTSP (10 to 1000 decision variables).

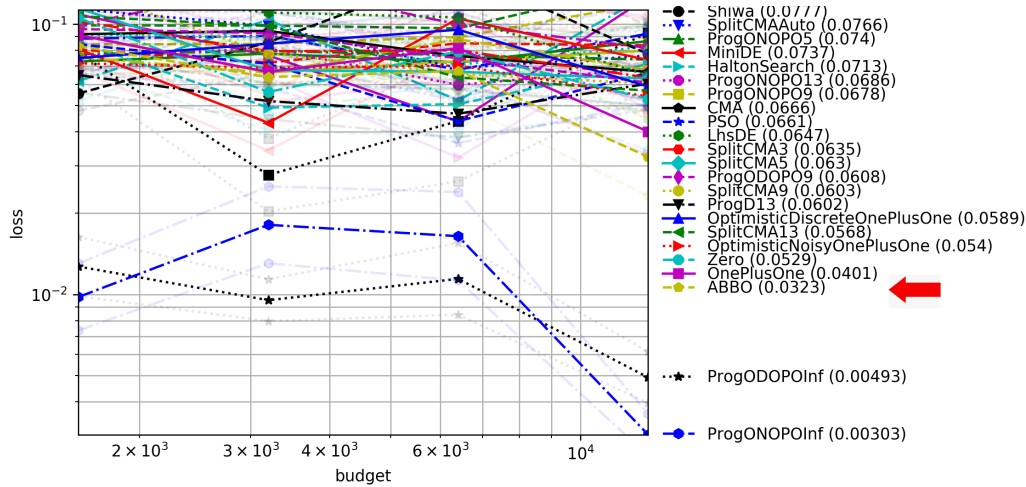

PowerSystems

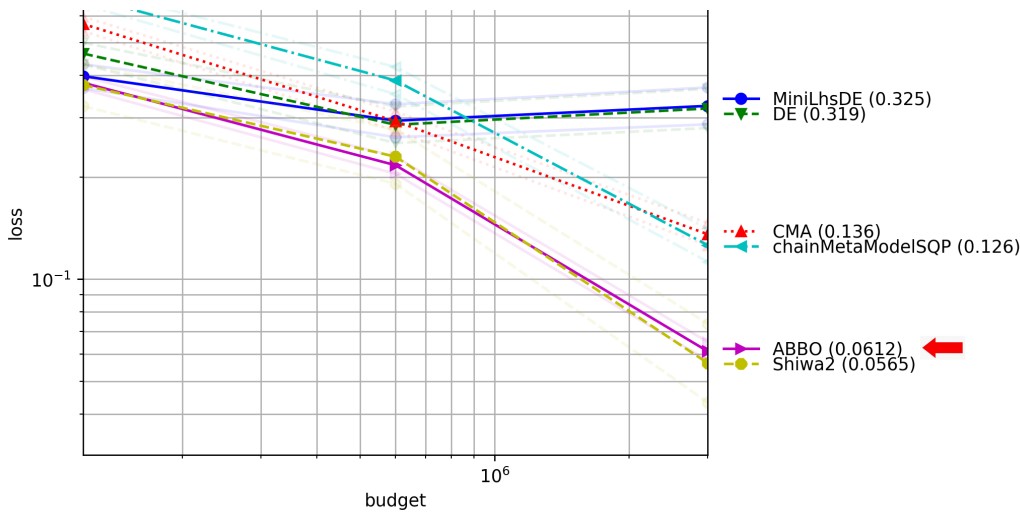

Lsgo (15 functions)

Figure 8: Additional problems (2): PowerSystems (1806 to 9646 neural decision variables) and LSGO (mix of partially separable, overlapping, shifted cases as in Li et al. (2013)).

