# OpenReview forum: "Black-Box Optimization Revisited: Improving Algorithm Selection Wizards through Massive Benchmarking"
_ICLR.cc/2021/Conference — Reject_

### Official Review · AnonReviewer3 · 2020-10-20
**Solid paper with valuable contributions**

**Rating:** 9
**Confidence:** 5

**Review:**

SUMMARY

The contribution of this work is two-fold: it collects an extremely wide range of benchmark problems for black-box optimization, and it proposes a new algorithm called ABBO. Both contributions are significant.

The proposed ABBO method is designed to be a swiss army knife solver, suitable for a wide range of different types of problems. To this end, naturally, it builds on existing components. Overall, the combination looks extremely convincing to me.


CRITICISM

Already the first sentence of the abstract is problematic:
"Existing studies in black-box optimization suffer from low generalizability, caused by a typically selective choice of problem instances used for training and testing different optimization algorithms."
This statement is very general. However, black-box optimization is an extremely wide area, ranging (at least) from mathematical optimization over evolutionary computation all the way to machine learning. The statement applied to various degrees to most studies in various subfields. I completely agree with the statement only when restricted to machine learning papers, where experiments are typically limited to very few RL benchmarks. In other areas things are far from perfect, but generally much better (it is understood that avoiding a bias completely is near impossible), since benchmarks e.g. with (YA)BBOB aim at general insights, not (only) at demonstrating peak performance. Please qualify this statement accordingly.

There is one more problematic statement in the abstract:
"A single algorithm therefore performed best on these three important benchmarks, without any task-specific parametrization."
Well, this "single algorithm" is really an algorithm selection machine. Technically it is "a single algorithm", but it is much more useful to think of ABBO as a selection and configuration method. In my understanding this makes a big difference. For optimal performance we need both: powerful components and powerful configurators. Please make absolutely clear that this "single algorithm" really is a configurator, which encompasses multiple components.

Personally I object some of the methods forming the basis of "Algorithm 1", although overall the choices look very solid. [Side note: I very much like the use of Powell's algorithms for fine tuning of approximate solutions found with more robust methods.] I have one suggestion: Diagonal CMA-ES is an outdated method. Please consider low-rank approaches as an alternative, like LM-CMA-ES, VD-CMA-ES and LM-MA-ES.
@inproceedings{loshchilov2014computationally,
  title={A computationally efficient limited memory CMA-ES for large scale optimization},
  author={Loshchilov, Ilya},
  booktitle={Proceedings of the 2014 Annual Conference on Genetic and Evolutionary Computation},
  pages={397--404},
  year={2014}
}
@inproceedings{akimoto2016projection,
  title={Projection-based restricted covariance matrix adaptation for high dimension},
  author={Akimoto, Youhei and Hansen, Nikolaus},
  booktitle={Proceedings of the Genetic and Evolutionary Computation Conference 2016},
  pages={197--204},
  year={2016}
}
@article{loshchilov2018large,
  title={Large scale black-box optimization by limited-memory matrix adaptation},
  author={Loshchilov, Ilya and Glasmachers, Tobias and Beyer, Hans-Georg},
  journal={IEEE Transactions on Evolutionary Computation},
  volume={23},
  number={2},
  pages={353--358},
  year={2018},
  publisher={IEEE}
}

I appreciate that the code is available. I understand that you start out by forking nevergrad. However, that's not a viable long-term strategy, at least when thinking in terms of utility for a wider community. Please put effort into merging your (im my opinion very significant) contributions back into nevergrad.

Algorithm configuration and selection for optimization is not entirely new. In the bbcomp results, the AS-AC-CMA-ES by Nacim Belkhir seems to be a very successful competitor method. I do not know whether the code is available or not -- I found Nacim's profile on github, but no code base corresponding to his competition entries. If possible, it would be very interesting to compare to his results.

This brings me to one of the few weak points of the paper. Experiments are performed for ABBO, but the authors rely (solely) on the nevergrad leader board for comparing with competitors. This has pros and cons. The huge advantage: results are not biased by running competitor methods with sub-optimal parameters. This is a huge plus; in effect, this is a rare case where I fully trust all experimental results. However, this means that some interesting baselines may be missing, in particular methods that predate nevergrad (which is still rather new), like AS-AC-CMA-ES.


MINOR POINTS

The fonts in all plots in figures 2 to 5 are far too small, in particular when printed. On screen I need to zoom in quite a bit. I understand that there are space constraints, but in this form the presentation of the results is of limited value.

I really do not understand why the machine learning community keeps talking about losses (and sometimes regrets) when it comes to optimization. The term "fitness" in evolutionary computation is no better. A long-established terminology exists already: the thing we minimize is an "objective function", and its value at a specific point is an "objective value". I vote for paying more attention to using the standard terminology (in general, not only in this paper), since it is compatible across multiple sub-communities of optimization.

Last paragraph of section 2: "Rocket" is listed twice.

It seems that some of the URLs in the references do not work (any more). The bbcomp website has moved here: https://www.ini.rub.de/PEOPLE/glasmtbl/projects/bbcomp/index.html
I did not find a replacement for the Artelys link, but maybe referencing the bbcomp results does the job.


RECOMMENDATION

Overall this is a very nice and valuable paper with two significant contributions. I strongly recommend to accept the paper.

---

> ### Author Response · Authors · 2020-11-15
> **Answer to reviewer 3**
>
> Abstract 1: we modify the sentence accordingly, pointing out that it applies to machine learning papers.
> We agree that there are many areas in which the statement should be different. We do believe, however, that OptimSuite is one of the biggest, in particular in terms of integrating machine learning (with MuJoCo, other RL tasks, Keras hyperparameter tuning, Scikit hyperparameter tuning, black-box adversarial attacks).
>
> Abstract 2: We agree that the statement should be different. We modify accordingly. It is important to acknowledge the importance of base algorithms and stay modest.
>
> References: we add these references. A new variant of CMA was recently added but none of those ones, and there is a github issue suggesting this. We agree that these 3 new variants are important.
>
> Merging into Nevergrad: We do it. A large part of OptimSuite is actually already merged, and the rest is already in PR under peer reviewing for code quality. As you can see in the code, there are many other new benchmarks under addition.
>
> Nacim's work: We completely agree that AS-AC-CMA-ES is a very interesting part of the state of the art. However, his point is a bit different:
> He works on per instance configuration, using a part of the budget for computing features. Our work is only based on (i) offline choice based on dimension / type of variables / budget / parallelism and such features available a priori (ii) chaining (running several algorithms in a row) (iii) bet and run, i.e. run several algorithms during e.g. 10% of the budget and then keep only the best.
> We work on continuous, discrete, noisy, etc forms of optimization.
> Therefore, this work is more complementary than comparable. What we should do is to include AS-AC-CMA-ES as one of our base components. We add this comment in the paper.
>
> Comparison with only algorithms from Nevergrad: Agree. However we believe that Nevergrad has one of the biggest set of algorithms and the list is growing. HyperOpt has just been added, DEAP is under work, new discrete evolutionary algorithms were also added, as well as NSGA2 in the multiobjective case. The set of algorithms is ever growing, and our interface makes it easy to contribute new algorithms.
>
> Minor points: Action: we entirely revisit the presentation (using the additional allowed page), add text overviews of the figures. Our results are also available in the leaderboard. We remove all occurrences of “fitness” and “loss”  in the paper. We keep regret as it has a specific implication (translation by the optimum value when it is known, and expected value after corruption by noise in the noisy setting) . Other suggestions have been taken into account.

---

> > ### Comment · AnonReviewer3 · 2020-11-15
> > **Thanks a lot!**
> >
> > Thank you for the reply. I think that your project makes an extremely important contribution, even more to machine learning than to optimization. Maybe not ABBO as a method will make a big difference, but the testbed you created could have an extremely positive long-term impact. In its current state the biggest danger to the field of machine learning is the reproducibility crisis. Work like yours is the most convincing counter measure I have encountered to date. Keep up the great work!

---

> > > ### Author Response · Authors · 2020-11-15
> > > **Thank you**
> > >
> > > Thank you very much for the encouraging words! And for the constructive comments. An updated version is now available.

---

### Official Review · AnonReviewer1 · 2020-10-22
**Lacking novelty and relevance to the ICLR community**

**Rating:** 5
**Confidence:** 4

**Review:**


** Summary **

This paper proposes a new benchmark suite for black box optimization algorithms, which contains a mixture of existing tasks such as nevergrad functions and RL tasks from MuJoCo. In addition, the authors propose a new automated approach to algorithm selection (ABBO), which uses a series of rules to select the best method.

** Primary Reason for Score**

The benchmark suite is predominantly an amalgamation of existing tasks, and only appears to be an incremental improvement vs. Nevergrad. The proposed meta-algorithm is built on a set of “hand-crafted selection rules” (in the authors’ own words). This does not seem like a meaningful contribution to the ICLR community, but may be interesting in other, more applied venues.

** Strengths **

1) The proposed OptimSuite may be more convenient for users.
2) The rules-based system works surprisingly well, which begs the question of whether a learned version could perform better, which may be of interest to the ICLR community.

POST REBUTTAL:
3) Including all benchmarks into one codebase could improve reproducibility.

** Weaknesses **

1) I don’t think ICLR is the right venue for this work, which seems more engineering focused and may be more suited to an applied venue.
2) Most of the benchmarks in this suite are all included elsewhere already. Many recent papers proposing blackbox optimization algorithms include several of them. In particular, using Nevergrad + MuJoCo seems to achieve most of the desirable properties.
3) The selection wizard is just a set of heuristics/hand-engineered rules. This might be more useful for industrial applications, rather than ICLR.
4) Given that this paper is about benchmarking optimization algorithms, the presentation of the results is poor. It is almost impossible to read the plots, and there is no central table/comparison of the different methods. Reading this I learn very about the algorithms that are being benchmarked.
5) I have several issues with the presentation of the RL results:
a) ARS is not even a blackbox optimization algorithm, it uses information about the MDP for state and reward normalization (v2-t).
b) The “SOTA” results have been copy and pasted from another paper, which pasted them from the ARS paper, which referenced 2017 results. These are nowhere near “SOTA with grad” in RL. In fact, ARS didn’t say they were SOTA, it simply said that was what TRPO got as a baseline. SOTA now is probably MBPO or SAC/TD3, all of which solve these tasks in a fraction of the time.
c) Missing relevant literature: there have been several approaches to blackbox optimization for RL (and other functions) since ARS. I included ones presented either at ICLR or similar venues, which should be discussed:
  i) **Gradientless Descent: High-Dimensional Zeroth-Order Optimization**. Daniel Golovin, John Karro, Greg Kochanski, Chansoo Lee, Xingyou Song, Qiuyi (Richard) Zhang. *ICLR 2020*.
  ii)  **Learning to Guide Random Search**. Ozan Sener, Vladlen Koltun. *ICLR 2020*.
  iii)  **From Complexity to Simplicity: Adaptive ES-Active Subspaces for Blackbox Optimization**. Krzysztof M. Choromanski, Aldo Pacchiano, Jack Parker-Holder, Yunhao Tang, Vikas Sindhwani. *NeurIPS 2019*.

---

> ### Author Response · Authors · 2020-11-15
> **Answer to reviewer 1**
>
> Learnt rules: this work is in progress. It is useful in terms of getting rid of human contributions and more principled thresholds in the rules. We provide in the dashboard the CSV files of our results, so that anyone can do experiments without running anything: it is in particular possible to run experiments based on training rules on some benchmarks and validating them on another benchmark.
>
> (1) Relevance for ICLR: we believe in high quality benchmarking for increasing the quality of publications. Our paper contradicts a previous NeurIPS 2020 paper claiming poor performance of CMA or Shiwa on MuJoCo. In our platform we immediately get good results for those algorithms. BBOB and LSGO probably contributed a lot for improving the quality of scientific publication in the evolutionary computation community: we believe our framework goes one step beyond by integrating real-world and machine learning and by unifying many benchmarks. Since the time of submission we have added Keras problems and Scikit-Learn tuning in OptimSuite (see updated PDF for details). The fact that our paper incidentally actually outperforms previous publications on black-box MuJoCo says a lot. Using OptimSuite, in contrast, reduces the possibility for deriving and publishing misleading or biased results. In addition, our setup enforces reproducibility of experiments. We also point out other random unexpected scientific outcomes from this big experimental comparison: the surprising effectiveness of Powell’s algorithm, once its asymptotic regime is reached; the effectiveness of Softmax for transforming a discrete optimization problem into a noisy continuous one; and the relevance of bandits methods for the most challenging continuous cases.
>
>
> (2) Benchmarks already available elsewhere: this is not the case for all benchmarks. In addition, this is a lot of existing benchmarks with a common interface. In addition, we fix various issues. For example, two codes are available for LSGO: we found differences between the two, and follow the correct one. Also, Noisy-BBOB has known drawbacks as discussed in the paper and in various posts on internet: our version YANOISYBBOB corrects these flaws. We have also an interface to Pyomo and a long list of problems: we believe this is the only framework with so many benchmarks. In addition, we have a wide range of optimization methods readily available, and we do recompute periodically all benchmarks.
>
> (3) ABBO: we point out that such rules are now the standard procedure in combinatorial optimization and planning: competitions are won by such methods (except of those which explicitly forbid combinations of preexisting works, such as the SAT competitions). In addition, this actually provides a lot of rigor: the poor results of CMA or Shiwa in the LAMCTS paper published at NeurIPS are probably due to a bad interfacing.
>
> (4) We increase the size of figures for improving the readability. We add arrows specifying where is ABBO in the plots. For additional readability, please note that Nevergrad’s dashboard is publicly visible online. It contains an explicit ranking for every benchmark. It is available at https://dl.fbaipublicfiles.com/nevergrad/allxps/list.html
> The paper is not only about benchmarking: we believe that alg. selection should become, in black-box optimization, as central as in combinatorial optim.
>
> (5) We agree that your references are relevant and thank the reviewer for them, but we point out that they either skip some of the MuJoCo tests or use an instance-specific hyperparameter tuning which questions its generality. One of the papers with instance specific HP tuning presents results clearly better than ours. However, the entire budget of the grid search should be included, which would make the figures so much different. In addition, parts of the code are not public.
> Papers (i) and (iii) did not evaluate their methods on the high dimensional problems “Ant” (dim 888) and “Humanoid” (dim 6392): they are the two most difficult problems. The paper (ii) reached the targets on all environments with a very limited budget in appearance. We tried to reproduce their results using the github repo. 1st, a part of the code (gradient variance reduction), is not OS. 2nd, it comes out that a grid search was performed to reach the performance announced in the papers, hence biasing the results. Moreover, the optimal HP are not specified in the paper, and seemingly a different tuning was used for each problem with per-problem choice of HP. Using the default HP given in the repo, we reached the target for Half-Cheetah with a budget ~56k: far higher than our results. For Humanoid with HP from ARS-v2, the target is reached at ~768k -- higher than our max-budget 500k. The reproducibility of our work is one of its strengths since there are no tuning of HP and runs are publicly made available.

---

> > ### Comment · AnonReviewer1 · 2020-11-15
> > **Placeholder**
> >
> > Thank you for your detailed response! I will go through this in detail and come back shortly.

---

> > > ### Author Response · Authors · 2020-11-15
> > > **Revised version now uploaded**
> > >
> > > Dear reviewer, thanks for the swift acknowledgement that you have seen our answers. The revised version is now also uploaded. Please do not hesitate to post further comments or questions. We thank you for your efforts in reviewing this paper, and for the constructive comments.

---

> > > > ### Comment · AnonReviewer1 · 2020-11-16
> > > > **Still not convinced of relevance + presentation issues.**
> > > >
> > > > ** The Good News
> > > >
> > > > I am increasing my score by one, because of the following:
> > > > 1) Having read through all the conversation, I realize didn't appreciate enough the reproducibility gains from amalgamating existing environments.
> > > > 2) The added ML hyperparameter tuning experiments (Keras and sklearn) make the work more relevant for ICLR.
> > > >
> > > > ** Remaining Issues (referencing above)
> > > >
> > > > 1) Relevance to ICLR: I still do not think ICLR is the right venue for this work. For instance, not a single previous ICLR paper is cited, the most relevant work all seems to be at GECCO. For example, ABBO is a direct follow-up to work that was published at GECCO.
> > > >
> > > > 2) Highlighting new tasks: I appreciate the highlighting of which benchmarks are new. It is hard to know how impactful this is though. For instance, if someone were to publish a new blackbox paper, would they necessarily use these? Of course they're nice to have, but I suspect most people will still just use Nevergrad + MuJoCo.
> > > >
> > > > 3) ABBO handcrafted: The arguments provided further reinforce my belief this is useful for industry/practical settings, but not for ICLR.
> > > >
> > > > 4) Plots not clear: The arrows do help. However, the results still aren't clear/easy to read. Maybe it would be better to have a big table instead of Fig 2,3,4 and then put those plots in the appendix 2-4x the size they are now.
> > > >
> > > > 5) RL Experiments: I suggest changing the language of the RL experiments, the results presented for gradient based are not at all state of the art. "SOTA with grad" is completely wrong. This was pasted from a 2017 paper (indirectly from other pasting). It is possible this could be fixed with communication. For example, I think for a *linear policy* some of these results would be SOTA specifically for that setting. But then you shouldn't compare against TRPO-nn, because once you open that door you should also compare against TD3/SAC (two very obvious RL algorithms that get ~3x the reward you get on HalfCheetah in significantly fewer timesteps). So it should be phrased as "blackbox optimization of a *linear policy* for RL" and then remove the gradient stuff, it isn't relevant, unless it was run on the same architecture. Another thing is that this is quite low dimensional. It would be interesting to see how these blackbox algorithms compare for optimizing larger neural network policies, as was included in the OpenAI ES paper (Salimans 2017).
> > > >
> > > > 6) Which acquisition function was used for the BO baselines? It seems the comment that it "explores the boundaries first" is highly dependent on the setup. A greedier acquisition function like UCB will probably not do that. I feel the BO comparison is a little weak, in that no recent methods have been compared (for example TuRBO, Eriksson et al NeurIPS 2019).

---

> > > > > ### Author Response · Authors · 2020-11-17
> > > > > **answer to reviewer 1**
> > > > >
> > > > > (new version of the paper to be submitted soon)
> > > > >
> > > > >
> > > > > Relevance: We have added reference to ICLR & NeurIPS papers, about MuJoCo. We include several machine learning tasks which are new in Nevergrad (MuJoCo, Keras, Scikit-Learn, 007). ABBO significantly outperforms Shiwa which was published at GECCO and is shown in our plots.
> > > > > Hyperparameter tuning and reinforcement learning are usual in machine learning conferences. LMRS was published at ICLR. We claim the best really entirely open sourced results on MuJoCo: in addition our work makes it possible for people to play with MuJoCo by uploading their code if they do not have the license, as Nevergrad periodically reruns everything and our work is integrated in Nevergard.
> > > > > Our references include 3 ICML, 3 NeurIPS, 1 ICLR (it is true that there was no ICLR paper before your comment). We do believe that the fact that this work combines references from mathematical programming and evolutionary computation and uses a standard platform from derivative-free optimization is a strength.
> > > > >
> > > > > Highlighting new task and risk of competing with Nevergrad:
> > > > > We totally agree that it is preferable for our community to have one common interface. This is why we have chosen to integrate all our work in Nevergrad. This way, the functionalities are identical, the algorithms from Nevergrad can be used, and we maintain all strong features of Nevergrad. Please note that our work is a temporary fork of Nevergrad. A large part of OptimSuite has already been merged in Nevergrad, and the remainder is currently under code review. Our fork will be deleted soon, we do not want to compete with Nevergrad: we want to contribute to it. The fact that MuJoCo (and all the other suites mentioned in the paper) are now accessible through Nevergrad is one of our contributions - they were not there before the present work.
> > > > >
> > > > > ABBO handcrafted: We agree that an automatic learning would be more elegant. This is not yet ready, unfortunately. In terms of interpretability, however, we point out that rules might be more interpretable than something automatically learnt on traces of previous runs.
> > > > > We point out that the fully automated nature of ABBO has a strong advantage: we do not run the benchmark plenty of times for tuning hyperparameters for each task specifically. When the dashboard of Nevergrad is created, all algorithms are run, with the same HP - or if the algorithms do something for modifying the HP, their evaluations for doing so is counted in the budget.
> > > > > Algorithm selection is standard in other areas of optimization: we believe it should become standard in black-box optimization.
> > > > >
> > > > > Unclear plots & appendices: Thanks: we will modify this accordingly. We will upload a new version of the paper in the next days.
> > > > >
> > > > > Mujoco experiments: We agree that this table is misleading. We modify accordingly. Yes, we work in the linear policy setting. We will make it clear our comparisons are for Linear policies in the blackbox setting.
> > > > >
> > > > > Bayesian optimization tools in Nevergrad: We use the Bayesian optimisation chosen by Nevergrad. It is the package https://pypi.org/project/bayesian-optimization/. It uses UCB by default (https://github.com/facebookresearch/nevergrad/blob/master/nevergrad/optimization/optimizerlib.py#L1552). We add the reference. Recently HyperOpt has been added. We agree that Turbo would be a great addition. We also add the reference. Importantly, we do not claim that Bayesian optimization does not work -- the results are just those of one Bayesian optimization library.

---

### Official Review · AnonReviewer4 · 2020-10-26
**A good paper but slightly below the acceptance threshold**

**Rating:** 7
**Confidence:** 3

**Review:**

The paper proposes a benchmarking suite to overcome the problem low generalizability with black box optimization algorithm. The benchmarking suite consists of standard academic benchmarks to real world optimization problems. It also covers several scenarios such as dynamic-static, small to large-scale, discrete to mixed-integer etc. This is a relevant contribution to the machine learning however there are several drawbacks which pushes back it's acceptance into ICLR.

(a) The state of the art discussion was good but why benchmarking is crucial and how it is implemented in Optimsuite was very short in description. That was supposed to be the main highlight of the paper whereas discussion in that part was not clear at all. Specifically, more descriptions were needed in terms of what features to include/exclude in the benchmarking suite and how that helps in generalizability.

(b) I know author(s) mentioned about interpretability as future work however I felt really challenging to understand the benchmarking suite especially when you have a combination of academic benchmarks and real world optimization problems together. I think that's a very significant challenge in implementing ABBO.

(c) One thing was not clear to me how the tuning parameters that are often associated with several optimization problems are handled here? are you keeping the tuning parameter same across all the competing methods?

---

> ### Author Response · Authors · 2020-11-15
> **Answer to reviewer 4**
>
> (a) why benchmarking is crucial: the example of MuJoCo shows how much a correct interfacing of benchmark with optimization methods is important. The results of CMA or Shiwa are excellent in our paper, whereas the results of CMA and Shiwa are poor in the LAMCTS paper and the setting is quite similar: in addition, we have no task-specific parameter tuning, and even no hyperparemeter tuning at all on the test benchmarks (Section 4.2 presents test benchmarks, not used for designing ABBO), so that our results are very reproducible and unbiased. We have a common interface for all methods and all benchmarks. Also, many papers use per-instance hyperparameter tuning: we have a single method, aggregating many base methods, for many benchmarks. This makes a difference as the massive cost of hyperparameter tuning is typically not displayed in experimental results of previous papers. More on this in answers to other reviewers.
>
> What to include in OptimSuite (which is basically the new version of Nevergrad): Our main goal is to have a diverse set of benchmark problems and instances in our test suite, with the ambition to cover many different scenarios met in ML. OptimSuite provides a unique interface to these. However, we also need to balance execution times, accessibility of the benchmarks, quality of the available baselines, difficulty of interfacing the suites with nevergrad, etc. Thus, all in all, our choices were to a large extent driven by human expertise. We also emphasize that we do not consider the design process to be finished. New benchmarks may be added in the future. However, please note that the current selection already makes a huge step forward, compared to the isolated benchmark suites that are currently studied and which hardly allow to compare solvers on more than one suite.
> A contribution of our work is that we fix issues where needed. For example, we found that the two implementations of LSGO differ: our version matches the one which appears to be correct. We have YABBOB as an analogous of BBOB, but we fix known issue of the noisy BBOB testbed (see paper for details and references). We include a correct interface to MuJoCo, so that everyone can run it -- and if you can’t run it for some reason, you can upload your algorithm and the platform will run it for you.
>
> Description: the code is entirely provided as an anonymized code. The experiments are fully described there, in readable Python and docstrings. A large part is already merged inside Nevergrad.
>
> (b) it is true that, at present, the interpretability of ABBO is questionable. We are working on extensions that automatically learn the rules from the dashboard. For the results reported in this paper, we have manually selected the constants, through expert guessing and trial/error on the training instances. We did not yet perform automated algorithm configuration on top. This will be done, but will require significant effort, given that we aim for very broadly applicable designs. We think providing a CSV file of performances (algorithm, problem) is a step forward to this direction.
>
> (c) We use all optimization algorithms implemented in Nevergrad as base methods. That is, for the time being, we do NOT perform hyper-parameter tuning nor do we modify the algorithms in any other way. Sometimes, several variants of a method are proposed, with different parameters (for example DE has many variants). The only thing we do is the list of rules so that ABBO redirects to the relevant optimization method, and a few chaining as mentioned in the description of ABBO. This was done by looking at results on the training benchmarks. We do not contribute to the tuning of each base method: we use the default as in Nevergrad.
>
> Modifications of the paper:
> - Develop justifications of why benchmarking is crucial and mention elements in (a) above.
> - Mention that we do not work on the parameters of the base methods -- this is prior work of contributors to Nevergrad.
> - Mention the learnt counterpart of ABBO as further work (already in progress).

---

> > ### Comment · AnonReviewer4 · 2020-11-15
> > **Happy with the author(s) response**
> >
> > Thanks very much for all the clarifications especially I appreciate the honest response provided in reply to my query about interpretability. I understand the challenges as highlighted and look forward to the future implementations to overcome it. Overall, great job in answering all the queries. The importance of benchmarking and how the choices of the hyper parameter are made is clear to me now.
> >
> > I changed my rating to "Accept".

---

> > > ### Author Response · Authors · 2020-11-15
> > > **Thank you**
> > >
> > > Thank you for the encouraging words and for revising the score. We also want to let you know that the revised version is now uploaded. Please do not hesitate to post further comments or questions. We thank you for your efforts in reviewing this paper, and for the constructive comments.

---

> > > > ### Comment · AnonReviewer4 · 2020-11-24
> > > > **Revised version looks good**
> > > >
> > > > Thanks very much for providing the revised manuscript! Looks good to me.

---

### Official Review · AnonReviewer2 · 2020-10-27
**Promising approach with issues**

**Rating:** 6
**Confidence:** 3

**Review:**

The paper proposes a benchmark suite for black-box optimization that covers more
different types of problems than existing benchmarks. They derive an algorithm
selection system for black-box optimization from it and evaluate its performance
empirically, comparing to other black-box optimization solvers.

The authors address an interesting problem and demonstrate some good empirical
results. It is certainly beneficial to have large benchmark suites to get a
comprehensive picture of the performance of different approaches.

While the general motivation for a new benchmark suite is clear, the specific
properties the authors list are not. Some of the properties in Table 1 seem to
be complementary (far-optimum and translation), while it is unclear why others
are important. It is certainly nice to have an automated dashboard (of what
exactly?) and one-line reproducibility, but within the general context of the
paper, which seems to focus more on generalizability of results, this seems
unimportant. Whether a particular set of benchmarks is complex is to some extent
a question of the definition of "complex", which the authors do not make clear.
I do not know what "human-excluded/client-server" means. While the majority of
properties listed in Table 1 are obviously important for black-box optimization,
it seems that the authors started from the properties that their benchmark suite
has.

The benchmarks themselves seem to be mostly existing benchmarks that were
combined into a new collection -- it would help if the authors pointed out what
benchmarks were specifically created for OptimSuite.

It is unclear how exactly ABBO was created -- was the set of rules determined
over all benchmark instances to maximize a performance metric? ABBO is listed as
one of the main contributions of the paper and it should be explained in more
detail how it was created and how it works.

The labels in the figures are too small.

Update after rebuttal period: Thank you for your clarifications. I have revised my score accordingly.

---

> ### Author Response · Authors · 2020-11-15
> **answer to reviewer 2**
>
> Far-optimum means that the algorithms have to go far from the center of the search space to find the optimum, whereas translation means that the optimum has been randomly shifted so that an algorithm can not “cheat” by being biased towards a specific part of the domain.
>
> Benchmarks:
> We distinguish in the paper benchmarks that were used for specifying rules in ABBO, and benchmarks which were reserved as a test set (e.g. power systems, mujoco, etc.): see the difference between 4.1 and 4.2. Instances of artificial benchmarks are randomized (random translations at least, and random rotations in some cases), so that even for benchmarks in the training set the exact instances used for producing plots were not used for the training.
>
> How ABBO was created: mostly by human expertise, but also by trial and error on the training benchmarks. The instances of the training artificial testbeds, and the instances of the training testbeds preserved for test were not used for creating the plots. Please see our answers below also for possible extensions towards automated learning and hyper-parameter optimization.
> For OptimSuite, the following benchmark suites were created by us for the present paper or significantly modified and were never used in previous papers: MLTuning, PowerSystems, SimpleTSP. For the following ones we created interfaces to existing platforms/codes: Rocket, Pyomo (with significant new pieces of code as other Pyomo codes are not used through black-box optimization). For LSGO, we recreated a code, exactly matching one of the existing platforms which is, according to us, the correct one. This is now also specified in the paper, at the end of Sec. 2.
>
> Modifications in the paper:
> - We increase the size of figures / labels.
> - We modify the explanation of far-optimum.
> - We explain how ABBO was created.
> - We mention which benchmarks were created specifically for OptimSuite, which is basically the new version of Nevergrad.

---

> > ### Comment · AnonReviewer2 · 2020-11-15
> > **Thank you.**
> >
> > Thank you for your reply!

---

> > > ### Author Response · Authors · 2020-11-15
> > > **revised pdf now uploaded**
> > >
> > > A short message to let you know that the revised version is now uploaded. Please do not hesitate to post further comments or questions. We thank you for your efforts in reviewing this paper, and for the constructive comments.

---

> > > > ### Comment · AnonReviewer2 · 2020-11-16
> > > > **Clarification on properties of benchmarks**
> > > >
> > > > I wonder if you could clarify on how you selected the desirable properties of benchmarks that you list. You have already addressed my question with respect to the far optimum, but it still seems to me that you can achieved this by translation as well, i.e. if translation is present, far-optimum isn't necessary.
> > > >
> > > > Can you also please comment on the other properties? I'm still not sure what some of them mean.

---

> > > > > ### Author Response · Authors · 2020-11-17
> > > > > **Properties of benchmarks**
> > > > >
> > > > > We hope that the following explanation clarifies the properties listed in Table 1. We write these developments in the appendix.
> > > > > - Large scale: includes dimension > 1000.
> > > > > - Translations: in unbounded continuous domains, a standard deviation has to be provided, for example for sampling the first and second iterates. Given a standard deviation std, we consider that there is translations when optimas are randomly translated by a N(0,std^2) shift. Only interesting for artificial cases.
> > > > > - Far-optimum: optima are translated far from the optimum, with standard deviation at least N(0, 25std^2).
> > > > > - Symmetrizations / rotations: rotation: with a random rotation matrix M, the function f can be replaced by RotatedF(x) = f(M(x)). Symmetrization: f(x) can be replaced by f(S(x)), with S(x) a diagonal matrix with each diagonal coefficient equal to 1 or -1 with probability 50% if the optimum is at 0.
> > > > > - One-line reproducibility: Where reproducibility requires significant coding, it is unlikely to be of great use outside of a very small set of specialists. One-line reproducibility is given when the effort to reproduce an entire experiment does not require more than the execution of a single line. We consider this to be an important feature.
> > > > > - Periodic automated dashboard: are algorithms re-run periodically on new problem instances? Some platforms do not collect the algorithms, and reproducibility is hence not given. An automated dashboard is convenient also because new problems can be added “on the go” without causing problems, as all algorithms will be executed on all these new problem instances. This feature addresses what we consider to be one of the biggest bottlenecks in the current benchmarking environments.
> > > > > - Complex or real-world: Real-world is self-explanatory; complex means a benchmark involving a complex simulator, even if it is not real world. MuJoCo is in the “complex” category.
> > > > > - Multimodal: whether the suite contains problems for which there are local optima which are not global optima.
> > > > > - Open sourced / no license: Are algorithms and benchmarks available under an open source agreement. BBOB does not collect algorithms, MuJoCo requires a license, LSGO and BBOB are not realworld, Mujoco requires a license, BBComp is no longer maintained, Nevergrad before OptimSuite did not include complex ML problems without license issue before our work: some people have applied Nevergrad to MuJoCo but with our work MuJoCo becomes part of Nevergrad so that people can upload their code in Nevergrad and it will be run on all benchmarks, including MuJoCo.
> > > > > - Ask/tell/recommend correctly implemented: The paper “On Object-Oriented Programming of Optimizer - Examples in Scilab” by Y. Colette et al. develops arguments in favor of the ask and tell format. The idea is that an optimization algorithm should not come under the format Optimizer.minimize(objective_function) because there are many settings in which this is not possible: you might think of agents optimizing concurrently their own part of an objective function, and problems of reentrance, or asynchronicity. All settings can be recovered from an ask/tell optimization method. This becomes widely used. However, as well known in the bandit literature (you can think of pure exploration bandits, as in Bubeck et al. 2009), it is necessary to distinguish ask, tell and recommend: the recommend method is the one which proposes an approximation of the optimum. Let us develop an example explaining why this matters: the domain is {1,2,3,4}, and we have a budget of 20. NoisyBBOB assumes that the optimum is found when “ask” returns the optimum arm: no matter what happens afterwards, the status remains "found" for ever as soon as the optimum has been asked. So an algorithm which just iteratively “asks” 1,2,3,4,1,2,3,4,.... reaches the optimum in at most 4 iterations. This does not mean anything, as the challenge is to figure out which of the four numbers is optimal. With a proper ask/tell/recommend, the optimizer chooses an arm at the end of the budget. A simple regret is then computed. Actually this also matters in the noise-free case, but the issue is more critical in noisy optimization. The case of continuous noisy optimization also has counter-examples and all the best noisy optimization algorithms use ask/tell/recommend.  We add the reference to the paper above.
> > > > > - Human excluded / client-server: The problem instances are truly black-box. Algorithms can only suggest points and observe function values, but neither the algorithm nor its designer have access to any other information about the problem apart from the number of variables, their type, ranges, and order. It is impossible to repeat experiments for tuning hyperparameters without “paying” the budget of the HP tuning. This is something we could not do, as everything is public and open sourced: however, we believe that we mitigate this issue by considering a large number of benchmarks.
> > > > >
> > > > > (new version of the paper to be submitted soon)

---

### Author Response · Authors · 2020-11-18
**New version uploaded**

Dear all,
We have just submitted a carefully revised version of our work. The revision addresses, in particular, the comments of Reviewers 1 and 2 (other comments were already addressed in our previous revision). We thank all reviewers for their constructive feedback, which has helped us to improve the presentation considerably. In particular, we agree that Table 2 is a nice way to aggregate the results from the plots (which are now available in the appendix).
We remain at your disposal for further discussions.
Best wishes,
the author(s)

---

### Decision · Program_Chairs · 2021-01-07
**Final Decision**

**Decision:**

Reject

**Comment:**

This paper presents a benchmarking suite, primarily targeting the domain of evolutionary style optimization algorithms, and an effective heuristic algorithm selection procedure ABBO.  The reviewers seemed quite split in their reviews with significant variance, particularly with one outlier review (9) lifting up the average.  They all felt that there was significant value in the work presented and that the benchmark could be useful for designing and evaluating new methods.  However, there were concerns regarding details about the contributions (e.g. a detailed description of ABBO and which contributions to the suite were novel vs obtained from other benchmarks), the relevance of this work to the ICLR community, and choice of algorithms presented (i.e. not SOTA).

In general, this seems like a useful contribution for the evolutionary algorithm community but this paper seems off-topic from the conference.  Certainly optimization is important and of interest to the community.  However, there is no machine learning component to the technical contribution of this paper, and it ignores many of the contributions to black-box optimization within this community (see e.g. the citations from AnonReviewer1, and the literature on surrogate-based black-box optimization - i.e. Bayesian optimization).  The RL optimization problems are somewhat relevant, but AnonReviewer1 raises concerns about the reporting of those results and the representation of the current literature.  There is an algorithm proposed in this work, but it's largely heuristic and no comparison is given to state-of-the-art portfolio optimization algorithms from the machine learning community (e.g. P3BO from Angermueller et al., ICML 2019).  A venue such as GECCO seems much more well suited to this work.